# Understanding Molecular Plant–Nematode Interactions to Develop Alternative Approaches for Nematode Control

**DOI:** 10.3390/plants11162141

**Published:** 2022-08-17

**Authors:** Mahfouz M. M. Abd-Elgawad

**Affiliations:** National Research Centre, Plant Pathology Department, El-Behooth St., Dokki, Giza 12622, Egypt; mahfouzian2000@yahoo.com

**Keywords:** plant–nematode interactions, nematode effectors and control, plant resistance

## Abstract

Developing control measures of plant-parasitic nematodes (PPNs) rank high as they cause big crop losses globally. The growing awareness of numerous unsafe chemical nematicides and the defects found in their alternatives are calling for rational molecular control of the nematodes. This control focuses on using genetically based plant resistance and exploiting molecular mechanisms underlying plant–nematode interactions. Rapid and significant advances in molecular techniques such as high-quality genome sequencing, interfering RNA (RNAi) and gene editing can offer a better grasp of these interactions. Efficient tools and resources emanating from such interactions are highlighted herein while issues in using them are summarized. Their revision clearly indicates the dire need to further upgrade knowledge about the mechanisms involved in host-specific susceptibility/resistance mediated by PPN effectors, resistance genes, or quantitative trait loci to boost their effective and sustainable use in economically important plant species. Therefore, it is suggested herein to employ the impacts of these techniques on a case-by-case basis. This will allow us to track and optimize PPN control according to the actual variables. It would enable us to precisely fix the factors governing the gene functions and expressions and combine them with other PPN control tactics into integrated management.

## 1. Introduction

With the ongoing nature of the socio-economic importance of agriculture, the global needs for sustainable and mounting food production to suffice the increased human population are evident. Thus, it is essential that issues associated with a full spectrum of crop production restrictions and losses are soundly solved. Plant–parasitic nematodes (PPNs) rank high among other crop pests and pathogens that constitute major constraints to agricultural production. Estimates of crop losses due to PPNs for the 20 life-sustaining crops averaged 12.6% of worldwide crop yield which equaled USD 215.77 billion of annual yield. An additional 20 crops with significant values for food and export have also a 14.45% annual yield loss which equaled USD 142.47 billion. The total 40 crops sustain an average of 13.5% losses which are estimated at USD 358.24 billion annually [1]. Clearly, these assessments will probably be elevated by adding other nematode-infected plant species worldwide to the list. Hence, adopting adequate and effective measures for optimizing PPN control tactics and strategies is a big challenge [2,3].

As PPNs are obligate parasites, they must feed on the roots or aerial parts of living plants to develop and reproduce either sexually or via parthenogenesis, i.e., reproduction without fertilization. While some PPN species have a restricted/narrow host range, others are polyphagous, but most PPNs are subterranean pests. Their second stage juvenile (J2) that hatches from the eggs is mostly the infective stage. It parasitizes plant roots of susceptible hosts, attracted to the roots via root exudates in certain host species. On the contrary, non-host/immune plants may have nematode-repellent materials in their roots. Hence, plant–nematode interactions have various aspects and may virtually occur even before the nematode touches the root of the plant (Figure 1). Eggs of *Globodera rostochiensis* need initial stimulation to hatch by chemical components in root exudates of their potato host [4]. Usually, many economically important and serious PPN species can attack and penetrate the plant roots and then migrate or develop feeding sites within them to secure their development and reproduction [5]. Therefore, a brief account of PPN categories is given in this review. Current PPN management measures to clarify their merits and demerits. Mounting concern about the demerits of PPN control methods has sparked broad interest in using resistant plant varieties/cultivars as safe, economic, and effective alternatives, especially to unsafe nematicides. This review addresses the mechanisms of natural resistance, especially against serious PPN species. It discusses current issues related to using both resistant plant genes and nematodes-effector proteins. These effectors have various functions, e.g., to detoxify enzymes in order to override the plant’s antimicrobial compounds, master host immune signaling, and keep their sound feeding structure as well as the essential processes for PPN development [6,7].

As a consequence of these issues, there is a desperate need to exploit modern molecular technologies in developing such alternatives for PPN control strategies. Grasping molecular plant–nematode interactions enables us to adopt crucial factors and harness efficient tools and resources for use in these strategies. Although a few molecular mechanisms of nematode infection during both plant–nematode compatible and incompatible interactions have already been explored with favorable results, many gaps are still there that pose difficulties in employing the related strategies [8,9,10,11,12,13]. Therefore, examples are given herein to clarify future proposed directions of various approaches based on boosting resistance in plants and/or suppressing nematode effectors. They address the most recent developments regarding the molecular basis underlying plant–nematode interactions with various techniques utilized to enhance plant protection against serious PPNs [13,14,15,16,17,18,19,20]. Their expansion in an integrated approach is presented to attain effective and durable employment in nematode management.

## 2. General PPN Categories and Management Measures

Above-ground nematode parasites are less abundant and comprise stem and bulb nematodes, seed gall nematodes, and foliar nematodes. For the subterranean phytonematodes, the mode of their parasitism may group them into four wide categories (Figure 1): (i) Ectoparasites, such as species of the genus *Helicotylenchus* and other spiral nematodes. They do not enter the roots but parasitize the root apex and/or peripheral cells via inserting long and robust feeding stylets. They usually move in soil searching for plant roots to parasitize. (ii) Migratory endoparasites, such as *Pratylenchus* spp. and *Ditylenchus dipsaci*. They can penetrate into and move within plant roots to feed and quit it to enter another one. After feeding, both ectoparasites and migratory endoparasites develop to J3, J4, and finally the adult without sexual dimorphism. (iii) Semi-endoparasites, such as *Tylenchulus semipenetrans* and *Rotylenchulus reniformis*, use only the nematode head to penetrate the root, but its posterior part remains in the soil. They are settled at one place on the root. The body of the female swells outside the roots. (iv) sedentary endoparasites, such as root-knot nematode (RKN) species (*Meloidogyne* spp.) and cyst nematode (CN) species (*Heterodera* and *Globodera* spp.). They enter and establish themselves within the roots. Both semi-endoparasites and sedentary endoparasites have sexual dimorphism. The swollen sedentary females sometimes protrude on the outside of the root. Migratory and sedentary endoparasites can damage plant tissues during their invasion, migration, and feeding on their susceptible hosts. Most of the studies on plant–nematode interactions have been centered on RKNs and CNs as their species are the most widespread and cause substantial crop losses in worldwide agricultural production. Launching and evolving of sedentary endoparasites-feeding sites for RKNs differ from that of CNs. The RKN J2 forms its feeding site on reaching the differentiating vascular tissue of plant roots via a few distinct giant/nurse cells, but cyst-forming J2 fixes it via setting syncytia close to the vascular bundle, where a few cells combine by resolving their cell walls. Having organized their feeding sites to transfer nutrients and solutes to the J2, the nematodes (RKN or CN) develop until reaching adult females via subsequent molts. These females lay eggs that hatch a new generation of J2s. Eventually, PPNs interact with their plant hosts in various courses ranging from transient ectoparasites to intimate involvement with their hosts, e.g., sedentary endoparasites.

Currently, PPNs are commonly managed via various production practices (chemical nematicides, bionematicides, resistant plants and crop rotation, soil amendments, fallowing, flooding, solarization, tillage, and use of certified transplants). Because most PPNs spend their lives within the soil or in plant roots, delivery of a chemical nematicide to the immediate surroundings of PPNs is generally difficult [21]. Yet, chemical nematicides are considered traditional means of effective PPN control (e.g., [22]). Unfortunately, the potential threat of these chemicals to wildlife, humans, and the environment, as well as the emergence of resistance-breaking nematode pathotypes/strains due to excessive use of these chemicals, has enforced the search for efficient and safe alternatives. Bionematicides are mostly safe alternatives, but they are frequently slower acting, less effective, and more inconsistent than these chemicals [23]. Using crop rotation is an effective and safe method for PPN control, were it not for the lack of PPN resistant/immune plant cultivars/varieties needed in the rotation. Soil amendments can enhance plant growth, but with the possible build-up in population densities of PPNs and BCAs, exceptions should be considered [24]. Related additions comprising botanical matrices and extracts, and purified secondary metabolites have received much research interest, but registration-processing and time-consuming issues have slowed their adoption [25]. Basic requirements for such materials are their safety, reliability, and favorable economics [26]. Fallowing and flooding may be used in PPN control but are not frequently economic for PPN control measures. Tillage is useful against many pests, weeds, and pathogens but can directly disrupt populations of PPN-antagonistic organisms and consequently increase nematode damage [24]. Certified transplants are excellent practices, but the plants should not grow into PPN-infested field soils, frequently an inevitable task. Eventually, the above-mentioned PPN chemical, cultural, and biological control techniques are not perfectly accepted and need deep revisions for safety and/or efficacy [19,27,28]. In addition, their demerits are frequently discouraging with regard to the generally low precision and accuracy in sampling the nematodes’ subterranean and within plant life stages as well as a wide host range of PPNs and their diverse and clumped distribution [29].

Hada et al. [18] have recently emphasized that it is difficult to recommend a favorable PPN management tactic that is reliable, economical, safe, and harmless to the nontargets. Rather, farmers and stakeholders would turn to resistant varieties/cultivars and production practices for PPN control, but, for numerous crops, these methods and resources are mostly unavailable or unfavorable. Grasping molecular plant–nematode interactions may offer novel approaches and resources to fill these gaps and assist in nematode control. If so, the related multiplex mechanisms, especially for sedentary endoparasites, regarding their feeding sites within the plant roots as well as cellular and sub-cellular responses in the PPNs and their host plants should be fully understood and exploited.

## 3. The Mechanism of Natural Resistance

Contrary to compatible nematode–plant interactions in susceptible hosts, the single dominant resistance genes from plants interact specifically with corresponding avirulence (Avr) genes in the nematode, leading to an incompatible interaction. This incompatible interaction commences a cascade of plant responses against the nematode—defense strategies. Plants experience several modes of action for protection and immunity. A general innate/basal immune system can recognize nematode-associated molecular patterns (NAMPs) by pattern recognition receptors (PRRs) as the primary defense line ‘layer’ against plant parasites. The extracellular receptor proteins (receptor-like kinases and receptor-like proteins) may be initiation factors to elicit basal immunity, e.g., against RKNs [9]. A conserved ascaroside (Asc#18) is reported as a NAMP of *Heterodera glycines* whereas the *Arabidopsis* leucine-rich repeat (LRR) receptor-like kinase NILR1 is yet the only known cyst nematode PRR. The activation of NAMP-triggered immunity (NTI) leads to a series of immune responses such as the production of reactive oxygen species and secondary metabolites, cell death around the PPN-migratory tract, and/or reinforcement of cell walls [30]. Such plant responses may decelerate the early stages of PPN infection. They can share in effective defenses but only in non-host plants. In nematode-susceptible plants, PPNs can overcome NTI via secreting effector proteins known as effector-triggered suppression (ETS) to inhibit the basal immune responses. Cyst nematode effectors, such as Ha18764, GrVAP1, RHA1B, and GrCEP12, are synthesized in nematode-esophageal glands and secreted into the roots by the PPN stylet [13]. Suppressing the innate immune responses usually results in establishing feeding sites (e.g., giant cells for RKNs and syncytia for CNs) necessary for nematode development and multiplication on their susceptible hosts.

Contrary to the first line or innate immune defense system, another defense line ‘layer’ is found only in PPN-resistant plant genotypes. A widespread thought is that it is encoded by single dominant resistance genes (*R*-genes) or quantitative trait loci (QTL) to manifest a host-specific defense [7,13]. Yet, the thought should be boosted by the fact that the *R*-gene may include a small gene family with highly homologous copies clustered together. Relevant intracellular signaling pathways must also exist to enable the expression of the resistant response. Although a single gene in the cluster may determine resistance, multi-gene families are common for plant *R*-genes. For instance, *Mi-1* comprises a small gene family with seven highly homologous copies clustered together on the short arm of chromosome 6 on resistant tomato [31], but several other *Mi* genes have been found, different from *Mi-1* in genetic locations, functional characteristics, and specificity [32,33]. Although ten genes are recognized for resistance to *Meloidogyne* spp. In tomatoes, only seven genes (*Mi-2*, *Mi-3*, *Mi-4*, *Mi-5*, *Mi-6*, *Mi-9*, and *MI-HT*) can operate at high temperatures, e.g., above 32 °C [9]. Likewise, for the cyst nematode, genome sequencing combined with fine mapping could indicate that the *H1* locus harbors a cluster of intracellular nucleotide-binding (NB)-LRR proteins (NLR) candidate genes, revealing that the *H1* gene is also a classical single dominant *R*-gene [34]. The most common class of intracellular proteins related to these *R*-genes usually encodes NLR to activate the host-specific defense.

The *R*-genes operate via two modes of nematode interaction. The first mode has a direct pathway relying on a direct gene-for-gene interaction where the receptor protein of the resistant plant interacts with the nematode effectors mastered by avirulence (Avr) genes. Striking output of this incompatible reaction is a localized programmed cell death so that no nematode-feeding site is formed. For instance, Avr genes of RKN produce effectors that trigger the production and the expression of plant *Mi*-resistant genes in tomato plants resulting in a type of hypersensitive response (HR) after the nematode enters the plant root [35]. Likewise, single dominant *R*-gene *H1* from potato plant can award resistance against avirulent *Globodera rostochiensis* populations [36]. The second mode of the host-specific defense is named the guard hypothesis. Its mechanism starts as nematode effectors trigger the plant-virulence factors (protein) which stimulates *R*-gene [37]. The Avr genes of nematodes interact with tomato accessory protein, for example, leading to some modification of this plant protein, enabling the recognition by plant nucleotide-binding site (NBS)-LRR proteins that monitor for infection. Consequently, RKN development is indirectly prohibited via inhibiting the formation of feeding sites.

While nematode effectors are intra- and extracellularly recognized by immune receptors, these latter, encoded by *R*-genes, have the same structural type as PRRs. However, enforced by *R*-genes, these PRRs can activate higher (specific) defense responses, upon direct/indirect recognition of apoplastic effectors produced by designated nematode strain(s) than defense by basal immunity. Yet, there are various modifications of the resistance mechanisms. While HR-induced resistance can cause necrosis of the nematode-feeding sites within two days post-infection, another resistance mechanism is not based on HR but rather disintegration of these sites at almost two weeks post-infection. This latter, the delayed disintegration of the feeding site, is noted in a broad variety of incompatible plant–nematode interactions, e.g., *M. incognita*-pepper, *H. schachtii*-sugar beet, *H. glycines*-soybean, *H. avenae*-cereals, and *Globodera* spp.-potato [38].

Even the same crop, such as pepper, may carry two resistance genes, *Me-3* and *Me-1,* for a quick HR soon after nematode inoculation and for delayed degradation of giant cells, respectively. Fewer RKN juveniles develop and reproduce on *Me-1* than *Me-3*-resistant hosts. Interestingly, a large number of resistance genes to RKNs are recorded to be located on the P9 chromosome of pepper [34]. Therefore, Abd-Elgawad [39] noted the importance of resistant pepper varieties as they can suppress RKN populations to low levels in soil with high fruit yield under high initial RKN pressure. Yet, careful manipulation of RKN resistance in pepper should be based on the fact that the resistance response is the result of the specific *R*-gene-*Meloidogyne* species and the plant genotype together. In other words, there are diverse mechanisms of resistance and therefore the plant defenses rely on activating many known and unknown *R*-genes or QTLs, especially for the economically impactful RKNs and CNs [14,15,40].

## 4. Successes and Difficulties in Using *R*-Genes 

Comprehensive references have addressed PPNs in temperate [41] and subtropical and tropical [5] agriculture materializing the successful use of naturally resistant plant species/cultivars. Although there are a good number of resistant genotypes, an urgent need is apparent for more ones to reduce PPN losses. Moreover, the majority of plant resistance genes used are effective against only the above-mentioned sedentary nematode category [30,33,42]. Hence, introgression of *R*-genes to confer nematode resistance to susceptible plants via classical genetic breeding can offer potent steps change in crop productivity [43,44,45]. Admittedly, plant genes responsible for PPN resistance are very useful in lowering PPN population levels, enhancing crop yields, and developing effective crop sequences.

In contrast to classical breeding for resistance, recognition and cloning of such genes found in a plant species can allow the transfer of resistance directly into other susceptible cultivar(s) with desirable traits of the same species, or even into cultivars of different species. Such genetic manipulations have the merits of avoiding linkage drag and scope to transfer resistance into genetic constitutions that prevent introgression by cross-breeding. Genes for nematode resistance could be cloned and transferred from some plant cultivars to others. The *Mi-1.2* from tomato against RKN (*Meloidogyne incognita*), *Hs1*^pro−1^ from *Beta procumbens* against beet CN (*Heterodera schachtii*), *Gpa-2* from potato against potato CN (*Globodera pallida*) and *Hero A* from tomato against potato CNs (*G*. *pallida* and *G*. *rostochiensis*) and *Cre* loci from *Aegilops* spp. against cereal CN (*H*. *avenae*) in wheat are apparent examples [28,46]. The arsenal of nematode-resistant genes, especially for major PPNs, still has additional favorable ones, e.g., *Me* in pepper, *Rk* in cowpea, *Rhg1* in soybean, *Ma* in *Prunus* spp., and *Mex1* in coffee. Their benefits may be exemplified in the enhanced resistance to RKNs that was achieved via cloning and transferring the full genomic region of the *Mi-1* gene found in tomato into a distant plant species, lettuce, *Lactuca sativa* [47].

Conversely, the lack of novel resources to back certain resistant plant species in controlling a few species of key nematode pests is consistently increasing due to the slow decline that could be noticed in their *R*-gene effectiveness. A remarkable example is the current problem of using resistance derived from plant introduction accession 88788 in 95–98% of the soybean cyst nematode (*H. glycines*)-resistant soybean varieties cultivated in the USA. Although *H. glycines* is the most important pest of the soybean there, the related plant resistance encoded by a high copy number of the *rhg1-b* allele has already started to decrease. Therefore, Kahn et al. [48] added *Bacillus thuringiensis* delta-endotoxin (Cry14Ab) as a plant-incorporated protectant. Consequently, genetically engineered soybean plants expressing Cry14Ab showed a decrease in *H. glycines* cyst and egg counts relative to control plants, demonstrating excellent potential of Cry14Ab to control PPNs in soybean. Another type of issue is related to the gene construct itself, e.g., single or dual genes. Tomato plants genetically engineered using double structure (PjCHI-1 and CeCPI) genes with synthetic promoters could generate transgenic lines that displayed a better decrease in RKN infection and reproduction than transgenic tomatoes with a single gene [49].

Additional cases are related to elements and components mediating *R*-genes. It is well established that salicylic acid (SA) and jasmonic acid (JA) can play a critical role in the signaling/expression of both innate and *R*-gene-mediated defense responses against pests and pathogens [50]. Remarkably, SA is involved in PPN-plant resistance, especially against sedentary forms. Therefore, the suppression of plant defense by PPNs is usually accompanied by the downregulation of the genes involved in SA-mediated defense. However, the SA-dependent pathogenesis-related protein genes *PR-1* (*P6*) were elevated rapidly in plant roots of susceptible tomatoes to levels comparable to that in resistant tomatoes; plants infected by *Globodera rostochiensis* showed similar free SA levels in the incompatible and compatible interactions [51]. Notwithstanding the utility of SA to enhance plant resistance, free SA levels in roots of infected susceptible plants may be impacted differently according to the attacking PPN species/genus. Molinari [7] speculated that the early and abundant necrosis caused by *G. rostochiensis* may trigger the noticed early but transient rise of SA with stimulation of SA signaling in susceptible tomato. Clearly, this level of stimulation for SA signaling does not occur in *Meloidogyne*-plant compatible interaction as RKN move intercellularly, causing less tissue damage.

Ultimately, plants can still be immunized against nematode attacks via pre-treatments with auxins that mediate defense reactions, e.g., SA. The beneficial rhizosphere microorganisms, such as arbuscular mycorrhizal fungi and biocontrol agents, e.g., *Trichoderma* spp., can induce systemic acquired resistance-like responses against RKN [52,53,54]. This does not negate the fact that more investigations on recognition/signaling pathways interacting with components or genes required for *R* functions are direly needed.

## 5. Common Issues of Natural Plant Resistance

### 5.1. Resistance Breaking Nematode Pathotypes

The development of resistance-breaking pathotypes has been extensively studied and reported (e.g., [9,33,55,56]). Although the above-mentioned selection pressure is a common cause to generate these pathotypes or virulent populations, an intriguing study [57] partitioned virulent RKN populations into (a) populations extracted from a field with grown resistant tomatoes, (b) natural virulent populations isolated from fields without grown resistant tomatoes, and (c) virulent populations selected from laboratory-avirulent populations. They concluded that the genetic events resulting in the acquisition of virulence against the *Mi*-gene differ between selected and natural virulent populations. Moreover, selection pressure for virulence could accompany gaining additional function enabling these PPNs to circumvent the host response, e.g., by enhancing antioxidant enzyme activities [58]. These virulent populations are becoming of wide occurrence [9]. Although they are especially found in monoculture systems which may support the selection pressure events, the exact reasons for their occurrence are unclear. It may also be due to ecological factors, e.g., temperature and changes in PPN populations. Ultimately, such virulent PPN populations, which can develop on resistant crops, would turn nematode resistance in sustainable agriculture into elusive strategies.

### 5.2. Genetics of Virulence in Nematodes

Certain nematode reproduction usually undergoes obligate mitotic parthenogenesis (i.e., *M. javanica*, *M. incognita*, and *M. arenaria*) in the tropics. Others, such as *M. chitwoodi*, *M. hapla*, and *M. fallax,* generally reproduce by facultative meiotic parthenogenesis in temperate climates. Cyst nematodes are largely amphimictic. Their species with facultative reproduction usually have a narrower host range than the asexual species. However, sexual reproduction boosts adaptability and heterogeneity among and within PPN populations [33]. Accordingly, virulent populations may be more inducible in those species of sexual multiplication. These populations were detected from avirulent strains too in resistant tomato fields with a monocropping system [9]. On the other hand, caution should be exercised for these virulent nematode populations, as it is well known that natural nematode resistance may be encoded not only by single dominant genes but also in a polygenic manner [33]. In this vein, sound use of statistics in nematology could be a helping tool. Therefore, high-quality sequencing and assembly via joining long-read sequencing to utilize high-density genetic mapping can boost the detection and characterization of PPN-virulent genes. This novel scheme can support our grasp of the plant–PPN interaction. 

### 5.3. The Temperature Factor

A remarkable example is the *Mi-1* gene of tomato used against RKNs. This gene cannot operate at temperatures above 28 °C for more than a few, maximum 48, hours after infection [35]. The RKN juveniles can establish their feeding site, relying on the temperature-dependent setback of resistance. Thereafter, resistance is not set any longer even at the permissive temperature. Therefore, HR-mediated resistance does not work to disrupt nematode growth and multiplication of the individuals that could form their feeding sites. Several factors were also reported to overcome *Mi-1*-mediated resistance. Populations of *M*. *javanica* and *M*. *incognita* that can infect and reproduce on tomato plants carrying *Mi-1* were documented [59]. Moreover, high population levels of *M*. *incognita* can seriously affect the resistance of the *Mi-1* gene [60]. On the contrary, *Mi-9*-mediated resistance is operating at high temperatures and is localized to the short arm of chromosome 6 of tomato [32]. Temperature is a pivotal factor as it impacts tomato resistance and the metabolic and PPN multiplication rates.

### 5.4. Improper Research Methods and Tools 

There are some molecular methods that should be dealt with carefully because they are based on materials that may be suitable for controlling a specific nematode genus but not others. Therefore, more studies with adequate tools and updated methods may be preferably directed towards nature, and structure of PPN-feeding tubes, the nematode-derived compounds, and consequent plant responses involved in such plant–nematode interactions for determining the molecular efficacy against the target nematode genus/species [9,10,12,16,27,61]. In this respect, as PPNs have a stylet orifice while feeding; it acts as a molecular sieve to uptake certain molecules and exclude others while feeding on tomato roots that express a nematicidal *Bacillus thuringiensis* crystal protein. The ultrastructure of these feeding tubes revealed that RKNs, but not CNs, can ingest larger transgenic proteins [62,63]. Thus, transgenic 54 kDa Cry6A and Cry5B proteins were ingested by and negatively affected *M*. *incognita* reproduction in tomato hairy roots [63,64]. On the contrary, resistance to cyst nematodes in roots expressing Cry5B protein from *Bacillus thuringiensis* is not conferred, i.e., the large 54 kDa Cry6A protein could not be ingested by *H*. *schachtii* due to the narrow orifice of the feeding tube; its size is limited to about 23 kDa [65]. This restriction severely limits the use of transgenic Cry proteins against some serious CNs.

Until not so long ago, there were many defects and flaws—now somewhat reduced—in the molecular tools and devices used. Remarkably, the quantitative polymerase chain reaction (qPCR) is superior to the frequently used PCR as the former enables not only the qualitative detection of target PPNs but also their quantification. It could be a faster and better alternative to the longstanding use of microscopy in PPN identification and counting during the study of nematode–host interactions, especially in developing countries. Although various qPCR diagnostic assays have been developed based on the internal transcribed spacer (ITS) of rDNA in many PPN species, related defects may arise. For instance, the high variability of ITS sequences in *Pratylenchus* spp. could enhance the risk of getting false-positive reactions (fragments from unidentified species) or false-negative reactions (variation existing between individuals). Moreover, imprecise quantification might also occur as some gene sequences are found in multiple copies in individual cells [66]. Furthermore, gene copy numbers can vary not only from one species to another but also amongst different PPN developmental stages [29]. Such confusing data may contribute to obtaining imprecise or unsound molecular nematode–host relationships. The main limit of qPCR is due to its failure to detect species that do not match the used primer/probes. Alternately, metagenomic methods can offer a reliable device, whether a PPN is found in databases, e.g., Genbank [67]. Based on the merits/demerits of each method, researchers should decide the approach that fulfills the intended goal(s).

Iqbal et al. [68] reviewed RNA interference (RNAi) of PPN genes as a now-common method. It involves engineering host plants to generate tall hairpin RNAs matching essential PPN genes. These genes are then processed into short interfering RNAs (siRNA) that trigger silencing as nematodes feed on cytoplasmic contents of the target plants [69]. They emphasized that the delivery of double-stranded RNA (dsRNA) to PPNs via host-induced gene silencing is more practical than spraying or any other method for a nematode-control strategy. Furthermore, they found that many of the tested genes reacted to RNAi knockdown differently [68]. Thus, they suggested that the original goal, types, R phenotypes of PPN strains, and current integration merits of RNAi should further be addressed; presumably, something more complex is occurring.

Common methods for transcriptome analysis of sedentary nematodes may rely on either isolating the nematodes from the plant tissue prior to RNA-sequencing (RNA-seq) or using dual RNA-seq where the plant roots and their invading nematodes are sequenced at the same time. The latter technique could have the merit of enabling PPN effector gene discovery and comparing the transcriptomic datasets between pre-parasitic and parasitic *Meloidogyne chitwoodi* juveniles on potato [62]. Thus, the dual RNA-seq could produce a substantial analysis of *M*. *chitwoodi* genes expressed during parasitism and encoded foreseen secreted proteins. This technique also considerably reduced the large list of genes in the *M*. *chitwoodi* secretome reported by the former method [70], isolating the nematodes from the roots led to recording genes not related to parasitism. While it is really difficult to functionally characterize ≥ 300 genes via a traditional method [70], dual RNA-seq could analyze the expression of fewer genes specifically at the early parasitic life stages of *M*. *chitwoodi* too [61].

## 6. General Approaches to Solve the Related Issues 

Basically, genetic improvement of plants for nematode resistance to enhance their productivity via traditional breeding or genetic engineering is likely only if the desired alleles are present in the gene pools of the targeted plants. A notable example of RKN resistance in tomato is that all its current resistant varieties originated from just the *Mi* gene. Resistance resulted from hybridizing the wild tomato plant (as a single resistance gene source) with the commercial one [9]. Breeders and stakeholders have worked on enhancing the effectiveness of resistant strains. Some of the related genes could work at high temperatures, e.g., *Mi-HT*, *Mi-2*, *Mi-4*, *Mi-3*, *Mi-5*, *Mi-6*, and *Mi-9* are heat stable. Yet, further surveys of other diverse habitats may find new and indigenous PPN-resistance genes—*R*-genes that do not rely on *Mi*-genes. 

Optimizing strategies for the efficient employment of durable resistant crops also requires a good knowledge of population genetics. As heat-stable resistance gene *Mi-9* is found in *Solannum arcanum*, resistance genes pyramiding in commercial varieties and genetic adjustments might enhance resistance durability. This could be done via manipulating plant metabolites that may comprise phenols, amino acids, and lipophilic molecules [71]. Furthermore, there is still much to grasp regarding resistance gene expression and function for various plant species and under different environments. Because there is great specificity of the virulent nematodes to the *R*-gene on which they were selected, the gene transfer or priming plants for immunization to counteract this virulence should be done using adequate molecular methods [7,33]. Moreover, durability could possibly be maintained via transferring multiple resistance genes to specific cultivar(s) within integrated nematode management systems. In such systems, using crop rotation and/or safe chemical nematicides can assist in reducing pressure on resistant cultivars/varieties to alleviate the emergence of virulent populations. BCAs can also offer a significant contribution to at least some of these systems. *Trichoderma asperellum* T34 reduced the number of eggs per plant of the virulent *M*. *incognita* population in both resistant and susceptible tomato cultivars. Fortunately, this fungal impact was additive with the *Mi-1.2* resistance gene of tomato [72]. Cloning and overexpressing the genes responsible for the biocontrol process from *Paecilomyces javanicus* may reinforce the plant immune response against RKN infection [16]. Likewise, engineered nanomaterials could show promising physical and chemical characteristics against nematodes [73].

Admittedly, examining the related biochemical, histological, and physiological aspects of plant–nematode interactions using sophisticated tools and devices may lead to novel and effective PPN management tools. A clear aim is to grasp molecular regulatory processes underlying PPN parasitism that could result in developing reliable PPN control strategies based on nematode genetic and plant-resistant backgrounds. In this respect, both the comprehensive secretome (different molecular proteins secreted via the nematode stylet that is repeatedly thrust into the cells of the plant roots) profiles and the whole-genome sequence of economically important PPN species have attained significant progress for important PPN species. For example, high-quality genome sequences of serious PPN species such as major RKN species [14,74,75,76] as well as less distributed ones, e.g., *Meloidogyne luci* [77], *M*. *enterolobii* [78], *M*. *exigua* [79], *M*. *chitwoodi* [80], and *M*. *graminicola* [81] are now available. Their availability should be harnessed not only to facilitate better comparative studies and phylogenomics on the related species but also help to recognize genomic variabilities and their main role in adaptability against different environmental factors and plant hosts, via examining the functional genomics. In this respect, a whole-genome shotgun study could reveal the long-read-based high-quality assembly of *M. arenaria* that may open new avenues to identify virulence-related genes [75]. These genes are frequently found in repeat-rich or highly variable regions in the genome. At hand, genome and transcriptome datasets are helpful in characterizing various PPN effector proteins and other genes involved in nematode parasitism. Additionally, more knowledge is still accumulating about these effector proteins to elucidate their significant roles during the penetration and migration within tissues of their plant hosts as well as parasitism comprising the adequate formation and maintenance of their feeding sites (e.g., nurse or giant cells for RKNs and syncytia for CNs), and deactivation of defense responses by their susceptible hosts [9,12,27].

Clearly, comparative secretome analyses among PPN species/strains/isolates are being investigated. They can determine which molecules are critical in inducing specific aspects of the disease and governing nematode virulence in the host plants. Thus, specific genes involved in the RNA interference pathways of the PPN species could be correctly targeted for nematode control [68]. Furthermore, combinatorial silencing of more than one functional gene at the same time could be more effective in PPN control [18]. Additionally, RNAi technology is being addressed to define specific PPN effectors to adapt them for effective nematode pest control. For instance, four isolates of the pinewood nematode, *Bursaphelenchus xylophilus*, with different levels of nematode virulence were recently compared to distinguish virulence determinants. These determinants, highly secreted by virulent *B. xylophilus* isolates, comprised Bx-CAT1 and Bx-CAT2 (as two C1A family cysteine peptidases), Bx-lip1 (lipase), and Bx-GH30 (glycoside hydrolase family 30). To quantitatively assess these four determinants at the transcript level at three stages, i.e., pre-inoculation, 3 days after inoculation (dai), and 7 dai into pine seedlings. Shinya et al. [20] used real-time reverse-transcription polymerase chain reaction analysis. They recorded significantly higher transcript levels of Bx-GH30, Bx-CAT2, and Bx-CAT1 in virulent isolates than in avirulent isolates at both pre-inoculation and 3 dai. While Bx-GH30 candidate virulent factor caused cell death in the plant, Bx-CAT2 was occupied in supplying nutrients for fungal feeding through soaking-mediated RNA interference. Shinya et al. [20] concluded that Bx-GH30 and Bx-CAT2 participate in the isolate virulence on host trees and may be engaged in pine wilt disease. Such nematode effectors can subsequently render themselves as potential candidate genes for nematode management. In this respect, RNAi may be utilized as a cellular procedure to degrade messenger RNA (mRNA), which plays the main role in protein synthesizing and consequently gene function. Thus, targeting ‘candidate’ effector genes of PPN species that cause successful infection of the host plant using the RNAi strategy could adequately suppress the genes responsible for this success [9,17,18,19,27,82]. The RNAi approach, for example, was utilized to knock down related effector genes of *Meloidogyne incognita* (e.g., *msp-16*, *msp-33*, *msp-20*, *msp-24*, and *msp-18*) that normally interact with plant transcription factors to express key cell wall-degrading enzymes (CWDE). The phenotypic plant data indicated that RNAi caused suppression of the targeted genes with a transcriptional shift in CWDE genes of the nematode [83].

## 7. Approaches to Strengthen Molecular PPN Control

Three main genetic classes for plant protection against PPNs have been used in a historical sequence, with overlapping between them. Traditional plant breeding for PPN resistance has long been used [84]. It has undoubtedly been progressing via genetic engineering too. This latter, the second class, aims at the general insertion of genetic material into a host genome. The latest class aims at genome editing in which DNA is inserted, deleted, modified, or replaced in the PPN genome. It aims at inserting genes to site-specific locations [85]. Ibrahim et al. [16] reported four main techniques of gene editing in order to boost the global breeding of cultivars resistant to RKN in a broad range of crops, namely recombinase-mediated site-specific gene integration, homologous recombination-dependent gene targeting, nuclease-mediated site-specific genome modifications, and oligonucleotide-directed mutagenesis. These techniques are expected to contribute to the rapid progress in grasping the plant–nematode interaction mechanisms and consequently ameliorate plant resistance against nematodes.

Rajput et al. [86] reviewed the related technologies, viz., the clustered regularly interspaced short palindromic repeat (CRISPR)/CRISPR-associated protein (CRISPR/Cas), as a strong device for accurately targeted modification of almost all crops’ genomes to produce variation and expedite breeding plans. O’Halloran [87] provided a soft program, CRISPR-PN2, as a conclusive web-based stage that offers elastic use and control over the automated design of specific guide RNA sequences for CRISPR experiments in parasitic nematodes. The effective use of CRISPR/Cas9-directed genome editing in plant species has also been reviewed by Ibrahim et al. [16] in chickpea, the legume models *Medicago truncatula* and *Glycine max*. Its technology permits high-throughput gene editing at the genomic scale. The editing may assist in enhancing desired traits in plants with a restricted genetic pool and insufficient resistance sources.

### 7.1. Expanding the Use of Marker-Assisted Selection 

Basically, marker-assisted selection (MAS) refers to utilizing a binding pattern of linked molecular (DNA) markers in order to indirectly select the desirable plant phenotype. Molecular markers are beneficial tools that can be used not only to set the introgression of genes related to economically desired traits but also to facilitate grasping molecular nematode–host interactions, as chromosome landmarks. Consequently, MAS are used for gene incorporation and stacking, as in tomato cultivars for multiple disease resistance traits. A striking example is *Mi-1* homologs that can grant resistance against a broad range of pests and pathogens, comprising the most common root-knot nematodes (*Meloidogyne javanica*, *M*. *incognita*, and *M*. *arenaria*), insects, i.e., potato aphids (*Macrosiphum euphorbiae*), and sweet potato whitefly (*Bemisia tabaci*), and oomycetes (*Phytophthora infestans*) in tomato plants [9,43]. Thus, various approaches relying on molecular markers for PPN resistance such as amplified fragment length polymorphisms (AFLPs), random amplified polymorphic DNA (RAPDs), restriction amplified length polymorphisms (RALPs), cleaved amplified polymorphic sequence (CAPS), reverse-transcription polymerase chain reaction (RT-PCR), single nucleotide polymorphisms (SNPs), sequenced characterized amplified regions (SCAR), sequence tagged site (STS), and simple sequence repeats (SSRs) are being developed [88,89]. They can be used to select a broad range of economically important plant species/cultivars for resistance against serious nematode pests (Table 1).

The readiness of marker application and affordability of marker genotyping, make MAS a good breeding option for many traits in most breeding programs. Moreover, marker development is advancing towards more reliable and efficient regeneration and genetic transformation systems with predictable and reproducible results. Very recently, the nematode resistance could be adequately addressed via using a genome-wide association study (GWAS) of single nucleotide polymorphism (SNP) markers with the PPN resistance. Thus, SNPs linked to resistance and the genes identified can establish a significant tool for introgression of resistance to *Heterodera glycines*, by marker-assisted selection in common bean (*Phaseolus vulgaris*) breeding programs [90]. Likewise, a locus on chromosome 13, comprising multiple TIR-NB-LRR genes and SNPs linked to *Meloidogyne javanica* resistance in soybean, was characterized by utilizing a combination of GWASs, resequencing, genetic mapping, and expression profiling [91]. Such technological progress would authorize a better understanding of gene function and expression with possible accredit of accurate genetic adjustment for PPN control and crop improvement [89].

**Table 1 plants-11-02141-t001:** Examples of molecular markers for screening nematode resistance in main crops.

Crop	Nematode Species	Resistance Genes	Marker Type	References
Tomato	*Meloidogyne incognita*	*Mi 3*	RAPD and RFLP	[92]
Eggplant	*Meloidogyne javanica*	*Mi-1.2*	RT-PCR	[93]
Wheat	*Heterodera avenae*	*CreX* and *CreY*	SCAR	[94]
Pepper	*M. incognita*, *M. arenaria*, *M. javanica*	*Me_3_* and *Me_4_*	RAPD and AFLP	[95]
Potato	*Globodera rostochinensis*	*H1*	RFLP	[96]
Soybean	*Heterodera glycines*	*Rhg1* and *Rhg4*	SNPs	[97]
Cucumber	*M. javanica*	*mj*	AFLP	[98]
Cotton	*M. incognita*	*qMi-C14*	SSR	[99]
Cotton	*Rotylenchulus reniformis*	*Ren^ari^*	SSR	[100]
Peanut	*Meloidogyne arenaria*	*Rma*	CAPS, SSR, AFLP	[101]

### 7.2. Utilizing Proteinase Inhibitor Coding Genes

Proteinase inhibitors (PIs) can hinder the function of proteinases/proteases released by the nematodes. As PPNs invade plants, these PIs become active against all nematode proteinases; aspartic, cysteine, metalloproteinases, and serine. Ali et al. [27] reviewed various applications of PIs against PPNs. They emphasized that simultaneous use of different PIs could have an additive effect as it combines specificity with a broad range of resistance. Pyramiding genes of taro cystatin and fungal chitinase with a synthetic promoter could also increase resistance to RKNs in tomato [49]. These and similar approaches of combining more than one biocontrol measure/agent [27,102,103] can form featured bases for elevating transgenic plant resistance. Cystatins from various plant species rank high among other PIs in boosting nematode resistance in a variety of crops. 

Abd-Elgawad [104] affirmed the importance of cystatins in increasing the nematode resistance within a plan that can upgrade eggplant production. Njom et al. [105] examined cysteine proteinases of the papain family (CPs) that attack nematodes and identified their specific molecular target(s). They concluded that multiple cuticle targets for these proteinases are found which probably make nematode resistance to these novel CPs slow to evolve. Thus, PIs have a future as a promising molecular control method against PPNs.

### 7.3. Use of RNA Interference

RNA interference is triggered by double-stranded RNA (dsRNA) inside the cell to degrade mRNA, the key to protein synthesis, and hence nematode-gene function. Therefore, the technique basically serves as a significant and robust device to analyze gene function in nematodes. Three classes of PPN-specific genes are being utilized as targets for RNAi techniques. These are genes enabling PPN parasitism, PPN developmental genes, and housekeeping genes [16]. As a genetically based approach, RNAi application has various aspects for effective and integrated control of PPNs. Its use to control plant infection with multiple plant pathogens proved to be promising [16,106]. This does not negate that the successful trials were solely based on single gene silencing for PPN control [82,83]. However, PPNs can masterly use several genes for accomplishing a specific function [18,19]. The nematode effectors found and expressed in subventral gland cells have many genes that can serve in nematode management as they are involved in the related nematode activities, e.g., penetration migration, and feeding within plant tissue. About 37 putative *M. incognita* esophageal gland secretory genes have been reported [83,107]. Nematode neuropeptides also serve in related processes such as host recognition, infection, and reproduction. This adds merit to the simultaneous silencing of genes as a promising tool in the control of PPNs. A dual gene construct of cysteine PI and a fungal chitinase with a synthetic promoter in transgenic tomato plants demonstrated considerably more reduction in RKN infection and reproduction than plants transformed with an individual gene [49]. Moreover, three *M. incognita* effectors, *Mi-msp1*, *Mi-msp16*, and *Mi-msp20* as fusion cassettes-1 and two FMRFamide-like peptides, *Mi-flp14*, *Mi-flp18*, and *Mi-msp20* as fusion cassettes-2 were successfully combined as targets of RNAi for nematode management. Their quantitative expression showed a significant decrease in mRNA abundance of target genes in *M. incognita* females in transgenic *Nicotiana tabacum* plants. The constructs, fusion 1 and fusion 2, granted up to an 85% decrease in *M. incognita* reproduction [18].

### 7.4. Nematicidal Proteins

Anti-nematode proteins such as some antibodies, lectins, and Bt Cry proteins can inhibit PPN development in plants. Yet, their mechanisms of inhibition vary and should be harnessed to optimize PPN control. Toxic lectins can block nematode-intestinal function [108]. The mechanism displayed by the lectins is crucial since several lectins bind with glycans. Overexpression of a *Galanthus nivalis* agglutinin (GNA)-related lectin driven by cauliflower mosaic virus promoter (CaMV35S) is exploited to offer anti-nematode efficacy in plants such as potato, oilseed rape (*Brassica napus*), and Arabidopsis concerning CNs, RKNs, and *Pratylenchus* spp. [27]. Some antibodies, known as plantibodies, are effective against PPNs in compatible plant–nematode interactions. They can oppose the active PPN-secreted proteins. They could, for example, react with secreted products of *G. pallida* and adversely affect the movement and invasion of this species to potato roots [109]. *Bacillus thuringiensis* (Bt) toxins, known as Cry proteins, could directly reduce the *M. javanica* population on tomato roots by adding bacterial suspension or spore/crystal mixture [110] or indirectly induce resistance against *H. glycines* in transgenic soybean plants [48].

### 7.5. Chemodisruptive Peptides

Usually, PPNs use chemoreceptive neurons to approach their host plants or get away from their non-host. These neurons discern certain chemical stimuli for attacking the plants. Acetylcholinesterase (AChE) and/or nicotinic acetylcholine receptors are usually used for adequate operation of their nervous systems. A few peptides, at such low concentrations, can bind with these receptors and consequently disrupt the PPN ability of chemoreception by hindering their reaction to chemical signals [111]. A peptide secreted by transgenic potato plants could inhibit *G*. *pallida*-AchE resulting in the disorientation of attacking nematode. This led to a 52% reduction in the number of *G*. *pallida* females [112].

These peptides could also offer the prospect of an integrated nematode control strategy. A repellant peptide precisely directed at the sites of *G*. *pallida* invasion via a root tip-specific promoter from an Arabidopsis gene could be combined with the transgenic expression of a rice cystatin in potato to maintain a high degree of potato plant resistance against this CN [113]. Transgenic maize plants demonstrated good PPN control via combining digestive protease inhibitor cystatin with synthetic nematode repellent peptides [114,115]. Likewise, using chemo-disruptive peptides alone or integrated with cystatins into various plant species has been documented to show high levels of resistance against RKNs with a consequent increase in crop yields [27,116].

### 7.6. Employing Plant Resistance Mechanisms

Fundamentally, the above-mentioned two layers of plant-induced resistance and their related mechanisms against pests and pathogens should be fully employed. For instance, seeds or roots of some plants can release PPN-killing or repelling compounds in their exudates [3]. Proteomic methods detected 63 exuded proteins from soybean seeds, comprising a trypsin inhibitor, a β-1,3-glucanase, a lipoxygenase, a lectin, and a chitinase, all can contribute to plant defense. These exudates were able to suppress the hatching of *Meloidogyne incognita* eggs and to cause full mortality of the J2. Pretreatment of J2 with these exudates resulted in a 90% decrease in the gall number on plant roots [117]. While such findings should be exploited, caution should be exercised in other cases, e.g., a new chemo-attractant synthesized on *Arabidopsis* seeds could attract different RKN species to invade the freshly emerged seedling roots [118]. Phytohormones have significant roles in plant–nematode interactions since sedentary PPNs can alter auxin homeostasis via multiple strategies. Recent functional analyses indicated that PPNs have developed multiple approaches to manipulate indole-3-acetic acid (IAA) homeostasis to set an effective parasitic relation with their susceptible plants [119]. In contrast, the role of other hormones, such as salicylic acid, could also be exploited to boost plant resistance against PPNs [7,33].

As yet, more information needs to be generated for the related genes and defense mechanisms to encompass various aspects of host-specific resistance to optimize their efficacy and durability in field crops. Such information is expected to circumvent or overcome many of the above-mentioned issues causing a lack of developing sufficient nematode-resistant plant species/varieties. Otherwise, the two general schemes used to transfer a PPN-resistance gene, i.e., from one plant species to another or from a cultivar to another one within the same plant species, may face unexpected difficulties. For instance, the fact that backcrossing of an *Hs1*^pro−1^ as a CN-resistant genotype and a susceptible sugarbeet plant did not lead to a resistance phenotype in the next generations is still raising an unsolved case [13]. Trials to transfer the *Mi-1* gene to *Arabidopsis* or tobacco were also ineffectual [33]. Conversely, favorable transfers of *R*-genes in heterologous species with monogenic resistance may result in resistance-breaking field pathotypes due to the imposed selection pressure. Other issues are related to the transfer of multiple disease resistance traits. These types are reflected in genetically transformed plants that share mixed characteristics of resistance and susceptibility. An outstanding example is the *Mi-1* gene transfer to eggplant from tomato. It could considerably lower RKN reproduction, but aphid resistance, displayed by the same transferred gene in tomato, was not attained [93]. This may raise the question of possible pleiotropic effects on the gene expressions. Another case of *R*-gene transformation raised the low level of resistance in the transformed plants as a result of the evident dosage impact of the *R*-gene copy number [33]. For such an impact, some authors assumed that expression of the resistance is more effective in homozygous than heterozygous genotypes of the tomato *Mi-1.2* gene, but others found the opposite for both the tomato *Mi-1.2* and the pepper *Me3* genes when the *R*-gene was introduced into homogeneous genetic backgrounds [120]. The authors assumed that transposable elements have a role in the creation and maintenance of *R*-genes-containing clusters in solanaceous crops as these elements are correlated with both large-scale genomic rearrangements and these genomic clusters. For instance, the sequencing of the P9 chromosome of pepper (carrying the *Me* gene cluster) showed how genome expansion due to these elements and duplication results in the advent of novel genes and functions or ‘neofunctionalisation’. The frequent clustering of *R*-genes may ease the harmony of plant defenses against simultaneous pathogenic species and the development of new specificities to target an ever-changing array of pathogens [120,121].

Admittedly, the genetic background into which these genes are introduced is of supreme significance to the expression of PPN resistance and its durability, as shown with other pathosystems. Therefore, genetic constitutions of susceptible plants selected for the manipulations should possess an additional proper set of intracellular signaling pathways in order to employ the transferred *R*-gene(s) in proper resistance mechanisms. If so, molecular plant–nematode interactions can effectively serve this direction for other *R*-genes conferring PPN resistance for which the relevant signaling pathways are still insufficient. 

### 7.7. Related Molecular Tools

Computational tools, bioinformatics technology, sound statistical methodology [121], and availability of increased molecular databases would ease grasping of the various types of nematode parasitism as well as the gene proteins, and recognizing pathways probably involved in plant–nematode interactions. These facilities, backed by reference genomes and novel genomic tools, comprising genotyping-by-sequencing (GBS), bulked-segregant analysis combined with whole-genome resequencing (BSA-seq), genome-wide association study (GWAS), and genomic selection (GS), will rapidly progress molecular PPN control to a high rank among other control measurements. This molecular PPN control can preferably be comprehensive in terms of addressing the management of polyspecific nematode populations [9]. Furthermore, the resistance to multiple PPN species should preferably be transferred into cultivars with resistance to other important pests. Clearly, researchers of relevant disciplines should better approach and apply the positive trends and standardization that serve this type for molecular control of PPNs. In this respect, novel plans to optimize nematode sampling [122] and focusing on recently recognized roles and tools to get better findings in the nematode realm are direly needed [16,123]. Such strategies can develop robust pest management programs able to efficiently replace unsafe nematicides while blocking the above-mentioned defects in the other control measures. Thus, sound integrated PPN management will combine molecular control and cultivation practices that lead to sustainable and high crop production techniques to keep the *R*-genes and sustain their durability.

Eventually, the regulatory picture for relevant transgenic plants is unclear and will stay so for the near future, as the scientific and social consequences of their release are debated. Meanwhile, there are recent references explaining the importance of using such modern methods to control PPNs on important crops such as tomatoes [124], and potatoes [125]. Although approaches to transgenic PPN control can be categorized as operating on nematode targets, nematode–plant interface, and plant response [126], the three classes might be interrelated to achieve the eventual control goal. Expression quantitative trait locus (eQTL) mapping-based approach to identify interacting sets of hosts and pathogen genes may further address how the PPN species can set gene expression differently on the account of their host’s genotypes [127]. Moreover, on the nematode side, exploration of our current understanding of plant–pathogen molecular interactions and how they differ among different life strategies of various PPN genera and species should be further boosted. There are many putative control targets in the PPN-life cycles that can be exploited as illustrated by Perry and Moens [128]. Comparative genomics will upgrade our understanding of their parasitic strategies and lifestyles as well as the vulnerable life stages. Addressing incompatible nematode-host interactions is direly needed for crop species with limited availability of genetic and genomic resources, e.g., near-isogenic and mutant lines, completed genome and transcriptome sequences, and commercial full genome arrays [129]. Yet, practical use of this information for environmentally safe PPN management options is challenging.

Hence, favorable research avenues to overcome these difficulties should rationally forecast them to bring significant productivity to commercial agriculture. Ultimately, rational molecular control of the nematodes would be better integrated with other pest management measures to maximize crop production. Finally, the above-mentioned approaches should be integrated into PPN management in real time. For instance, there is no evidence that the novel strains of BCA have favorable traits without hard and tiring screening focusing on their virulence and versatility [130]. The alternate method is a directed search of mediums where BCAs will have had to develop the needed traits. This approach requires close academic–industry partnerships and a change in mindset away from the mold of using the traditional pesticide model to timely achieve tremendous strides.

In conclusion, PPNs are causing global crop losses while their classical control methods are not sufficient. Grasping the molecular basis of their interactions with plants can assist in developing new methods for PPN-molecular control for better nematode management. Omics technologies based on genes and proteins involved in the nematode activities and the plant resistance responses are key factors to evolve this control. Techniques such as next-generation sequencing of genomes and transcriptomes, RNAi, PIs, MAS, chemo-disruptive peptides, and genetic transformation systems with reproducible results must be available within PPN control strategies to enhance crop yields. Linking long-read sequencing to the use of high-density genetic mapping can also support the detection and characterization of PPN-virulent genes. Progress in computational biology, bioinformatics, and analyzing the omics large-scale data can efficiently boost these techniques to offer accurate recognition of components and pathways engaged in PPN parasitism and plant response. Updated genome-editing devices will serve classical plant breeding and precisely translate how gene actions are linked to phenotypic performances. Yet, these methods must be cautiously used to avoid unwanted effects such as PPN virulence and pleiotropic impacts on qualitative and quantitative crop yields. Definite issues such as those related to the transfer of multiple disease resistance traits, gene original construct, and specificity of the virulent nematodes to the *R*-gene may arise in particular cases. Therefore, employing the outcomes of these techniques on a case-by-case basis is proposed. This will enable us to monitor and improve PPN management according to the given variables. Eventually, molecular control methods of PPNs should be combined with other control tactics for integrated management as a way forward in crop protection/pest management.

## Figures and Tables

**Figure 1 plants-11-02141-f001:**
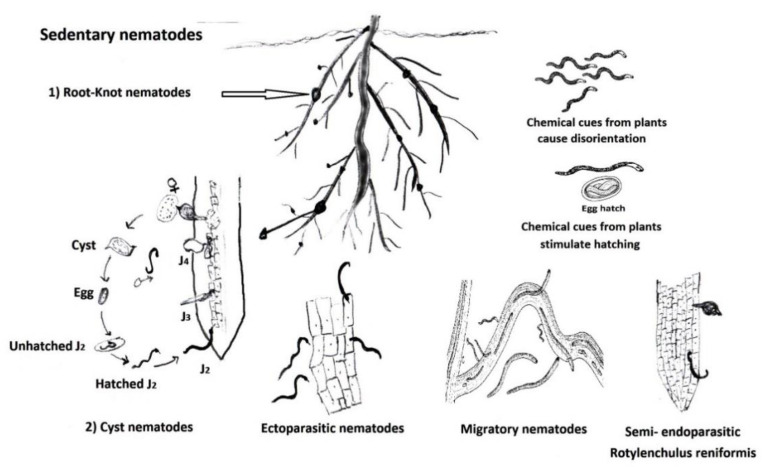
Categories of soil phytonematodes showing interactions with their host plants. Root metabolites can also function as PPN repellents, attractants, killers, or inhibitors.

## Data Availability

Not applicable.

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
