# Peer review of "Understanding Molecular Plant–Nematode Interactions to Develop Alternative Approaches for Nematode Control"

_plants, 2022, doi:10.3390/plants11162141_

Round 1
Reviewer 1 Report
The manuscript looks good to me. No further changes.
Author Response
Comment: The manuscript looks good to me. No further changes.
Response: Much appreciated.
Reviewer 2 Report
This manuscript reviews the current literature and knowledge regarding molecular plant-pathogen interactions and their application towards management of plant-parasitic nematodes in agriculture, particularly through host resistance. I think this is a worthwhile undertaking for a review manuscript and a scientifically interesting topic. The author gives good justification for the reason for needing a review article on this topic, and generally I agree with the author.
However, overall, the review is quite long and rambling in some areas, and thus could be shorted and condensed to the most important points. Some of the sections seem to jump around from topic to topic, or example to example quite frequently, making some parts difficult to follow. Other times, particular points made by the author are not fully explored. For example:
-Line 25: The reference to the Covid-19 pandemic does not add anything to the meaning of the manuscript.
- Lines 132-135: It is not clear how the "precision and accuracy of sampling", how the "wide host range", or how the "diverse and clumped distribution" of nematodes are "discouraging" to the use of chemical, cultural, and biological methods of nematode management?
- Lines 325-333: This section appears underdeveloped. There are very few citations in this section, too.
- Line 370-373: I'm not quite sure what these three sentences are trying to convey. They could be removed.
- Line2 492-496: It is not fully clear what these different techniques are and how they applied to study of host resistance genes and development of resistant cultivars, particularly for nematode management. It feels thought the thought was not fully finished.
- Line 620-622: Again, I'm not quite sure what these two lines are trying to convey. They could be shortened or removed.
There are also several places where the word choice is a bit strange or could be re-written for clarity. For example:
- Line 223: "familiar and non-familiar R-genes". Do you rather mean "known and unknown R genes" ?
- Line 223: "serious RKN and CN". How are nematodes serious? Do you mean "economically impactful RKN and CN"?
- Line 241: "Factually". All statements in a scientific manuscript should be factual, so stating that one is so is redundant. This can easily be removed.
- Line 292: "unpleasant fact expressed by several PPN resistance genes". This is a strange way to make this statement - resistance genes are not capable of "expressing" a fact, yet alone an "unpleasant one".
- These are just a couple examples throughout the manuscript. I think review of the text by a professional English editing service could assist with some of this.
Figure 1 is beautifully prepared and a nice illustration of the different feeding and life strategies of plant-parasitic nematodes. However, the exploration of our current understanding of plant-pathogen molecular interactions and how they differ among these different life strategies was a bit lost throughout the text. Also, I am not sure why Figure 1 is referenced in lines 603 or 630?
Thank you for the opportunity to review this manuscript. I think with some refocusing and rewriting of the manuscript, it could be quite impactful.
Author Response
Thanks for the third cycle of reviewer comments. Yet, I’d like to remind that this paper has been accepted before by “Plants” but the author has to make withdraw after its ACCEPTANCE due to unavailable APC. Now, the money is available but for LIMITED TIME AND FUND because a project that will end this August will fund it.
Thanks for the valuable reviewer’s comments and suggestions. The author complied with them as follows:
Comment: I think this is a worthwhile undertaking for a review manuscript and a scientifically interesting topic. The author gives good justification for the reason for needing a review article on this topic, and generally I agree with the author.
Response: Much appreciated.
Comment: However, overall, the review is quite long and rambling in some areas, and thus could be shorted and condensed to the most important points. Some of the sections seem to jump around from topic to topic, or example to example quite frequently, making some parts difficult to follow. Other times, particular points made by the author are not fully explored. For example:
-Line 25: The reference to the Covid-19 pandemic does not add anything to the meaning of the manuscript.
Response: Done, the reference to the Covid-19 pandemic was deleted.
Comment: Lines 132-135: It is not clear how the "precision and accuracy of sampling", how the "wide host range", or how the "diverse and clumped distribution" of nematodes are "discouraging" to the use of chemical, cultural, and biological methods of nematode management?
Response: Done, the sentence was re-written as follows to clarify it: “In addition, their demerits are frequently discouraging with regard to the generally low precision and accuracy in sampling the nematodes’ subterranean…”.
Comment: - Lines 325-333: This section appears underdeveloped. There are very few citations in this section, too.
Response: Done. This section was extended to include three more references as follows: “Several factors were also reported to overcome the Mi-1-mediated resistance. Populations of M. javanica and M. incognita that can infect and reproduce on tomato plants carrying Mi-1 were documented [59]. Also, high population levels of M. incognita can seriously affect the resistance of the Mi-1 gene [60]. On the contrary, Mi-9-mediated resistance is operating at high temperatures and is localized to the short arm of chromosome 6 of tomato [32]. Temperature is a pivotal factor as it impacts tomato resistance and the metabolic and PPN multiplication rates”.
Comment: - Line 370-373: I'm not quite sure what these three sentences are trying to convey. They could be removed.
Response: Done, the three sentences were deleted.
Comment: - Line2 492-496: It is not fully clear what these different techniques are and how they applied to study of host resistance genes and development of resistant cultivars, particularly for nematode management. It feels thought the thought was not fully finished.
Response: Detailed description of these techniques are found in the cited reference but their brief mention here points to their general contents. The reviewer is right and in order to fully finish the thought, the following was added “These techniques are expected to contribute to the rapid progress in grasping the plant-nematode interaction mechanisms and consequently ameliorate plant resistance against nematodes.”
Comment: - Line 620-622: Again, I'm not quite sure what these two lines are trying to convey. They could be shortened or removed.
Response: Done. This sentence was deleted “Although this theme is very important, it is reported herein as the last factor because it reflects the overall genetic background of plant-nematode interactions.”
Comment: There are also several places where the word choice is a bit strange or could be re-written for clarity. For example:
- Line 223: "familiar and non-familiar R-genes". Do you rather mean "known and unknown R genes" ?
Response: Thanks, ‘known and unknown’ replaced ‘familiar and non-familiar’
Comment: - Line 223: "serious RKN and CN". How are nematodes serious? Do you mean "economically impactful RKN and CN"?
Response: Thanks, ‘economically impactful RKN and CN’ replaced ‘serious RKN and CN’
Comment: - Line 241: "Factually". All statements in a scientific manuscript should be factual, so stating that one is so is redundant. This can easily be removed.
Response: Done, removed.
Comment: - Line 292: "unpleasant fact expressed by several PPN resistance genes". This is a strange way to make this statement - resistance genes are not capable of "expressing" a fact, yet alone an "unpleasant one".
Response: Done, instead, it is written ‘The development of resistance-breaking pathotypes has been extensively studied and reported (e.g., [9,33,55,56]).’
Comment: - These are just a couple examples throughout the manuscript. I think review of the text by a professional English editing service could assist with some of this.
Response: Dr. Zafar Handoo (working at Mycology & Nematology Genetic Diversity & Biology Laboratory USDA, ARS, Northeast Area, Bldg. 010A, Rm. 111, BARC-West, 10300 Baltimore Avenue, Beltsville, MD 20705, USA) has generously revised it.
Comment: Figure 1 is beautifully prepared and a nice illustration of the different feeding and life strategies of plant-parasitic nematodes. However, the exploration of our current understanding of plant-pathogen molecular interactions and how they differ among these different life strategies was a bit lost throughout the text. Also, I am not sure why Figure 1 is referenced in lines 603 or 630?
Response: In order to re-focus on these different life strategies that was a bit lost throughout the text, the following sentences, backed by a relevant reference, was added: ‘on the nematode side, exploration of our current understanding of plant-pathogen molecular interactions and how they differ among different life strategies of various PPN genera and species should be further boosted. There are many putative control targets in the PPN-life cycles that can be exploited as illustrated by Perry and Moens [128]. Comparative genomics will upgrade our understanding of their parasitic strategies and life styles as well as the vulnerable life stages. Addressing incompatible nematode-host interactions is direly needed for crop species with limited availability of genetic and genomic resources, e.g. near isogenic and mutant lines, completed genome and transcriptome sequences, and commercial full genome arrays [129]. Yet, practical use of this information to environmentally safe PPN management options is challenging..’ Also, referring to Figure 1 in lines 603 or 630 was deleted. The following new REF was added:
Perry, R.N.; Moens, M. Introduction to plant-parasitic nematodes; modes of parasitism. In Genomics and Molecular Genetics of Plant–Nematode Interactions; Jones, J.T., Gheysen, L., Fenoll, C., Eds.; Springer: Heidelberg, Germany, 2011; pp. 3–20.
Thanks again to the reviewer and I hope that my response to her/his comments is satisfactory as I did my best.
Reviewer 3 Report
It is interesting and well prepared review manuscript . I would recommend it for publication after a major revision.
Major comments
1. There are many nematology research groups working with molecular plant-nematode Interactions to develop effective control of nematodes. Unfortunately, author did not make references to some classical articles in this field published by Valeria M Williamson, Lieve Gheysen, Isgouhi Kaloshian and others. For example: J.P. McCarter, 2008. Molecular Approaches Toward Resistance to Plant-Parasitic Nematodes. Plant Cell Monogr, doi:10.1007/7089_2008_32 Isgouhi Kaloshian and and Marcella Teixeira (2019). Advances in Plant−Nematode Interactions with Emphasis on the Notorious Nematode Genus Meloidogyne. Phytopathology https://doi.org/10.1094/PHYTO-05-19-0163-IA
Isgouhi Kaloshian, Olivia J. Desmond, Hagop S. Atamian. Disease Resistance-Genes and Defense Responses During Incompatible Interactions. In: Genomics and Molecular Genetics of Plant-Nematode Interactions pp 309–324. I would strongly recommend to read all these publications as well as provide the list of reviews on this topic in the Introduction.
2. line 78-107. I would also recommend to use this reference to discuss the mode of parasitism - Chapter 1 Introduction to Plant-Parasitic Nematodes; Modes of Parasitism by Roland N. Perry and Maurice Moens. J. Jones et al. (eds.), Genomics and Molecular Genetics of Plant-Nematode Interactions,
DOI 10.1007/978-94-007-0434-3_1
Ectoparasites - Helicotylenchus and other spiral nematodes and endoparasites - Pratylenchus spp. , Ditylenchus dipsaci, not Scutellonema
Author Response
Thanks for the valuable reviewer’s comments and suggestions. The author complied with them as follows:
Comment: It is interesting and well prepared review manuscript. I would recommend it for publication after a major revision.
Major comments
- There are many nematology research groups working with molecular plant-nematode Interactions to develop effective control of nematodes. Unfortunately, author did not make references to some classical articles in this field published by Valeria M Williamson, Lieve Gheysen, Isgouhi Kaloshian and others. For example: J.P. McCarter, 2008. Molecular Approaches Toward Resistance to Plant-Parasitic Nematodes. Plant Cell Monogr, doi:10.1007/7089_2008_32 Isgouhi Kaloshian and and Marcella Teixeira (2019). Advances in Plant−Nematode Interactions with Emphasis on the Notorious Nematode Genus Meloidogyne. Phytopathology https://doi.org/10.1094/PHYTO-05-19-0163-IA
Isgouhi Kaloshian, Olivia J. Desmond, Hagop S. Atamian. Disease Resistance-Genes and Defense Responses During Incompatible Interactions. In: Genomics and Molecular Genetics of Plant-Nematode Interactions pp 309–324. I would strongly recommend to read all these publications as well as provide the list of reviews on this topic in the Introduction.
Response: Thanks a lot. The following references which cover significant contributions relevant to the topic are included with their related work in the text:
- Williamson, M.; Ho, J.Y.; Wu, F.F.; Miller, N.; Kaloshian, I. A PCR based marker tightly linked to the nematode resistance gene, Mi in tomato. Theoret. Appl. Gen. 1997, 87, 757–763.
- Goggin, L.; Jia, L.L.; Shah, G.; Hebert, S.; Williamson, V.M.; Ullman, D.E. Heterologous expression of the Mi-1.2 gene from tomato confers resistance against nematodes but not aphids in eggplant. Mol. Plant-Microbe Interact. 2006, 19, 383–388.
- Williamson, M.; Roberts, P.A. Mechanisms and genetics of resistance. In Root-Knot Nematodes; Perry, R.N., Moens, M., Starr, J.L., Eds.; CAB International: Wallingford, UK, 2009; pp. 301–325.
- Caromel, B.; Gebhardt, C. Breeding for nematode resistance: Use of genomic information. In Genomics and Molecular Genetics of Plant-Nematode Interactions; Jones, J., Gheysen, G., Fenoll, C., Eds.; Springer: Dordrecht, The Netherlands, 2011; pp. 465–492. https://doi.org/10.1007/978-94-007-0434-3_22.
- McCarter, J.P. Molecular approaches toward resistance to plant-parasitic nematodes. In Plant Cell Monographs: Cell Biology of Plant Nematode Parasitism; Berg, R.H., Taylor, C.G., ; Springer-Verlag: Berlin, Germany, 2008, pp. 239-268. doi:10.1007/7089_2008_32.
- Guo, ; Fudali, S.; Gimeno, J.; Digennaro, P.; Chang, S.; Williamson, V.M.; Bird, D.M.; Nielsen, D. Networks underpinning symbiosis revealed through cross-species eQTL mapping. Genetics 2017, 206, 2175–2184.
- Perry, R.N.; Moens, M. Introduction to plant-parasitic nematodes; modes of parasitism. In Genomics and Molecular Genetics of Plant–Nematode Interactions; Jones, J.T., Gheysen, L., Fenoll, C., Eds.; Springer: Heidelberg, Germany, 2011; pp. 3–20.
- Kaloshian, I.; Teixeira, M. Advances in plant−nematode interactions with emphasis on the notorious nematode genus Meloidogyne. Phytopathology 2019, 109, 1988-1996. https://doi.org/10.1094/PHYTO-05-19-0163-IA.
- Padilla-Hurtado, B.; et al. Evaluation of root-knot nematodes (Meloidogyne spp.) population density for disease resistance screening of tomato germplasm carrying the gene Mi-1. J. Agric. Res. 2022, 82(1), 157-166. http://dx.doi.org/10.4067/S0718-58392022000100157.
- Kaloshian, I.; Desmond, O.J.; Atamian, H.S. Disease resistance-genes and defense responses during incompatible interactions. In Genomics and Molecular Genetics of Plant-Nematode Interactions; Jones, J.T., Gheysen, L., Fenoll, C., Eds.; Springer: Heidelberg, Germany, 2011; 309–324. DOI 10.1007/978-94-007-0434-3_15.
For example, the following is added in the text: ‘Although approaches to transgenic PPN control can be categorized as operating on nematode targets, nematode-plant interface, and plant response [126], the three classes might be interrelated to achieve the eventual control goal. Expression quantitative trait locus (eQTL) mapping-based approach to identify interacting sets of hosts and pathogen genes may further address how the PPN species can set gene expression differently on the account of their host’s genotypes [127]. Also, on the nematode side, exploration of our current understanding of plant-pathogen molecular interactions and how they differ among different life strategies of various PPN genera and species should be further boosted. There are many putative control targets in the PPN-life cycles that can be exploited as illustrated by Perry and Moens [128]. Comparative genomics will upgrade our understanding of their parasitic strategies and life styles as well as the vulnerable life stages. Addressing incompatible nematode-host interactions is direly needed for crop species with limited availability of genetic and genomic resources, e.g. near isogenic and mutant lines, completed genome and transcriptome sequences, and commercial full genome arrays [129]. Yet, practical use of this information to environmentally safe PPN management options is challenging.’ Also, the following is added in the text: ‘Several factors were also reported to overcome the Mi-1-mediated resistance. Populations of M. javanica and M. incognita that can infect and reproduce on tomato plants carrying Mi-1 were documented [59]. Also, high population levels of M. incognita can seriously affect the resistance of the Mi-1 gene [60]. On the contrary, Mi-9-mediated resistance is operating at high temperatures and is localized to the short arm of chromosome 6 of tomato [32].’
Comment: 2. line 78-107. I would also recommend to use this reference to discuss the mode of parasitism - Chapter 1 Introduction to Plant-Parasitic Nematodes; Modes of Parasitism by Roland N. Perry and Maurice Moens. J. Jones et al. (eds.), Genomics and Molecular Genetics of Plant-Nematode Interactions, DOI 10.1007/978-94-007-0434-3_1
Response: I have actually added this reference referring to its valuable content, but I referred to their contribution as a means of solving the problem under a title ‘Related Molecular Tools’ and not as an introduction to it. The following is my addition ‘….Also, on the nematode side, exploration of our current understanding of plant-pathogen molecular interactions and how they differ among different life strategies of various PPN genera and species should be further boosted. There are many putative control targets in the PPN-life cycles that can be exploited as illustrated by Perry and Moens [128]. Comparative genomics will upgrade our understanding of their parasitic strategies and life styles as well as the vulnerable life stages.’ I hope it comply with your suggestion too.
Comment: Ectoparasites - Helicotylenchus and other spiral nematodes and endoparasites - Pratylenchus spp. , Ditylenchus dipsaci, not Scutellonema
Response: Done and corrected.
Thanks again to the reviewer and I hope that my response to her comments is satisfactory as I did my best.
Round 2
Reviewer 3 Report
Author addressed all comments and I believe that the manuscript could be accepted for publication.
This manuscript is a resubmission of an earlier submission. The following is a list of the peer review reports and author responses from that submission.
Round 1
Reviewer 1 Report
The review paper is well written and thorough. I just have a few minor comments:
Please make sure to add line numbers the next time. It would make it easier for reviewers to pinpoint issues. The received and accepted date mentioned on the side of the first page doesn't make sense. This paper has not been accepted yet. Please change the received date. The header on each page says Plants 2021, which means you have used the older version of the template. Please change it to 2022 and use the 2022 format template for this year for any future submissions.​ Citation for this sentence " Estimates of crop losses due to PPNs for the 20 life-sustaining crops averaged 12.6% of worldwidecrop yield which equaled $215.77 billion of annual yield" is missing. Change bases to basis in this sentence " They address the most recent developments regarding molecular bases underlying..." Remove the word 'so' in this sentence, "Hada et al. [18] have recently emphasized that it is so difficult...". It is unnecessary.
Remove italics from "Institutional review board statement, Informed consent statement, Data availability statement ".
Author Response
We thank the reviewer for his/her comments which will improve the manuscript. The responses to all the concerns that were raised are given below. We used the track system with another word color to indicate our complying with his/her comments in a supplementary manuscript.
Comment: Please make sure to add line numbers the next time. It would make it easier for reviewers to pinpoint issues. The received and accepted date mentioned on the side of the first page doesn't make sense. This paper has not been accepted yet.
Response: Line numbers have been added. The submitted version is a withdrawn manuscript after acceptance by “Plants” due to unavailable APS charges at the disposal of the author. Now money is available but for a relatively short time !.
Comment: Please change the received date. The header on each page says Plants 2021, which means you have used the older version of the template. Please change it to 2022 and use the 2022 format template for this year for any future submissions.​
Response: Done. However, any further change to suite the 2022 format template is the responsibility of the Journal team.
Comment: Citation for this sentence "Estimates of crop losses due to PPNs for the 20 life-sustaining crops averaged 12.6% of worldwide crop yield which equaled $215.77 billion of annual yield" is missing.
Rebuttal: It is cited as REF [1]. Because several successive sentences are closely related to the same point, the REF is cited at the last sentence.
Comment: Change bases to basis in this sentence" They address the most recent developments regarding molecular bases underlying..."
Response: Done.
Comment: Remove the word 'so' in this sentence, "Hada et al. [18] have recently emphasized that it is so difficult...". It is unnecessary.
Response: Done.
Comment: Remove italics from "Institutional review board statement, Informed consent statement, Data availability statement ".
Response: Done. Sincere thanks again for your constructive comments.
Reviewer 2 Report
The main aim of this work was to review the state of the art of the molecular plant-nematode interactions to be used as basis for plant-parasitic nematodes management. The most information of the manuscript corresponds to root-knot and cyst nematodes with some references to citrus, pine and lesion nematodes, consequently, the title should be changed. The manuscript is excessively long, with sub-sections that do not provide information relevant to the objective of the work, and in some cases with information that is not entirely accurate (for example, not all the plant-parasitic nematodes comprises the same number of life cycle stages), with some mistakes (for example, Me3 acts interfering with the formation of the feeding sites whilst the Me1 acts at nematode penetration into the roots), and without adequate references. The manuscript should be reorganized, shortened, with appropriate information in each sub-section, and centered into the objective of the manuscript.
Several omics studies have been carried out, some of them with susceptible and resistant plants and assessing both plant and nematode genes. Some others have been used mutant plants to determine the main factors involved in resistance. Surprisingly, none of them have been used in the review, or their findings have not been highlighted.
The temperature factor affects the expression of some plant resistance genes but not others. The lack of expression of R sensitive genes is reversible as the environmental temperatures decrease under a threshold.
I do not understand what is the "Improper reserach methods and tools" story.
The subsection concerning the selection of virulent nematode populations is not well solved, as well as the resistance durability. The example of the Mi9 gene is not representative. The Mi1.2 gene is affected by high temperatures, while Mi9 is not, but both are susceptible against nematode populations virulent to Mi1.2. Will the Mi9 gene be more durable under the current conditions of Mi1.2 use?
What about the plant-parasitic nematode community. R-genes are nematode specific and its use can lead to the increase of other components of the community. How the molecular plant-nematode interaction can aid with this issue?
The use of molecular markers can aid in plant breeding for resistance but the response of plants bearing the gene cannot be what be expected, as previously reported. What about that?
In my opinion, the manuscript is far from being considered as accepted for publication.
Author Response
We thank the reviewer for his/her comments which will improve the manuscript. The responses to all the concerns that were raised are given below. We used the track system with another word color to indicate our complying with his/her comments in a supplementary manuscript.
Comment: The main aim of this work was to review the state of the art of the molecular plant-nematode interactions to be used as basis for plant-parasitic nematodes management. The most information of the manuscript corresponds to root-knot and cyst nematodes with some references to citrus, pine and lesion nematodes, consequently, the title should be changed. The manuscript is excessively long, with sub-sections that do not provide information relevant to the objective of the work, and in some cases with information that is not entirely accurate (for example, not all the plant-parasitic nematodes comprises the same number of life cycle stages), with some mistakes (for example, Me3 acts interfering with the formation of the feeding sites whilst the Me1 acts at nematode penetration into the roots), and without adequate references. The manuscript should be reorganized, shortened, with appropriate information in each sub-section, and centered into the objective of the manuscript.
Rebuttal: In addition to the importance of the genera: root-knot and cyst nematodes (Meloidogyne, Heterodera, Globodera), each genus includes many nematode species that are widespread worldwide. We also mentioned other very important nematode genera related to the core of the research, such as citrus (Tylenchulus), lesion (Pratylenchus), and pine (Bursaphelenchus) nematodes. So I think the title is appropriate, as it is not possible to mention an extremely great number of plant-parasitic nematodes. More importantly, plant-parasitic nematode management or control mostly focuses on these genera due to their economic importance and worldwide spread. The goal of this manuscript is to address molecular approaches for nematode control. On the other hand, with such voluminous research and literature that are currently devoted to this topic, it is felt that the manuscript is not excessively long. I’d rather recall, for example, another reviewer’s statement that this manuscript addresses “a very present day issue in phytonematology and plant pathology, the plant-nematode interaction at cellular and molecular level. It is a very ambitious proposal.” I wonder because the reviewer’s statement that “plant-parasitic nematodes comprises the same number of life cycle stages” is not exist in the manuscript but the presented figure illustrates life cycle of definite nematode groups and genera. Likewise, the reviewer’s statement that “Me3 acts interfering with the formation of the feeding sites whilst the Me1 acts at nematode penetration into the roots” is not even implied in the context without due REF. On the contrary, the manuscript statement is that that “some authors assumed that expression of the resistance is more effective in homozygous than heterozygous genotypes of the tomato Mi-1.2 gene, but others found the opposite for both the tomato Mi-1.2 and the pepper Me3 genes when the R gene was introduced into homogeneous genetic backgrounds [120]” It is backed by the REF [120]. For reorganizing the manuscript with appropriate information: Factually, it is impossible to separate some points because they are tightly interrelated, so they may be briefly referred to again when they are closely interconnected with others. However, each title/subtitle occupies the largest and most important text related to it.
Comment: Several omics studies have been carried out, some of them with susceptible and resistant plants and assessing both plant and nematode genes. Some others have been used mutant plants to determine the main factors involved in resistance. Surprisingly, none of them have been used in the review, or their findings have not been highlighted.
Rebuttal: Given the fact that omics refers to a field of study in biological sciences that ends with -omics, such as genomics, transcriptomics, proteomics, or metabolomics, the manuscript has many direct or indirect related studies backed by REFs. They are focused on susceptible and resistant plants and assessing both plant and nematode genes. Examples are: “… should be harnessed not only to facilitate better comparative studies and phylogenomics on the related species but…”, “…plant hosts, via examining the functional genomics...”, “Omics technologies based on genes and proteins involved in the nematode activities and the plant resistance responses are key factors to evolve this control….”, “…analyzing the omics large-scale data can efficiently boost these techniques to offer accurate recognition of components and pathways engaged in PPN parasitism and plant response..”, “…cloning and transferring the full genomic region of Mi-1 gene found in tomato into a distant plant species, lettuce,..”.
Comment: The temperature factor affects the expression of some plant resistance genes but not others. The lack of expression of R sensitive genes is reversible as the environmental temperatures decrease under a threshold.
Response: Admittedly, the temperature factor affects the expression of some plant resistance genes but not others. The manuscript did not even imply otherwise. On the opposite, it was written that “Although ten genes are recognized for resistance to Meloidogyne spp. in tomatoes, only seven genes (Mi-2, Mi-3, Mi-4, Mi-5, Mi-6, Mi-9, and MI-HT) can operate at high temperatures, e.g., above 32 °C [9].” Because it is stated at the manuscript “temperature is a pivotal factor as it impacts tomato resistance and the metabolic and PPN multiplication rates”, so a separate section of the manuscript is devoted to “5.3. The Temperature Factor”. Eventually, the breakdown of Mi-1.2 resistance in tomato and the small number of major resistances genes currently available for root-knot nematodes should boost search for alternative strategies to develop durable nematode resistant crops; that is, not to be satisfied with reversible R sensitive genes. Because the paper addresses “Vital Alternative Approaches for Nematode Control” R sensitive genes were mentioned but not in such details.
Comment: I do not understand what is the "Improper reserach methods and tools" story.
Response: Numerous examples are given under this topic to present preferable methods/tools/materials versus less desirable ones with backing explanations and results (please see for example, qPCR vs. PCR; H. schachtii vs. M. incognita concerning ingestion of the large 54 kDa Cry6A protein, …etc.).
Comment: The subsection concerning the selection of virulent nematode populations is not well solved, as well as the resistance durability. The example of the Mi9 gene is not representative. The Mi1.2 gene is affected by high temperatures, while Mi9 is not, but both are susceptible against nematode populations virulent to Mi1.2. Will the Mi9 gene be more durable under the current conditions of Mi1.2 use?
Response: Initially, the manuscript addresses present pressing problems for effective PPN control via molecular approaches. The manuscript presented several such approaches that may solve the problems. For example, about the resistance durability, the manuscript stated: “Some of the related genes could work at high temperatures, e.g., Mi-HT, Mi-2, Mi-4, Mi-3, Mi-5, Mi-6, and Mi-9 are heat-stable”. For working against virulent nematode populations, the manuscript stated: “further surveys of other diverse habitats may find new and indigenous PPN-resistance genes—R genes that do not rely on Mi-genes.” Also, several approaches based on promising results or practical experimentation were reported as follows: “Optimizing strategies for efficient employment of durable resistant crops also requires a good knowledge of population genetics. As heat-stable resistance gene Mi-9 is found in Solannum arcanum, resistance genes pyramiding in commercial varieties and genetic adjustments might enhance resistance durability. This could be done via manipulating plant metabolites that may comprise phenols, amino acids, and lipophilic molecules [71]. Furthermore, there is still much to grasp regarding resistance gene expression and function for various plant species and under different environments. Because there is great specificity of the virulent nematodes to the R-gene on which they were selected, the gene transfer or priming plants for immunization to counteract this virulence should be done using adequate molecular methods [7,34]. Moreover, durability could possibly be maintained via transferring multiple resistance genes to specific cultivar(s) within integrated nematode management systems. In such systems, using crop rotation and/or safe chemical nematicides can assist in reducing pressure on resistant cultivars/varieties to alleviate emergence of virulent populations. BCAs can also offer a significant contribution to at least some of these systems. Trichoderma asperellum T34 reduced the number of eggs per plant of the virulent M. incognita population in both resistant and susceptible tomato cultivars. Fortunately, this fungal impact was additive with the Mi-1.2 resistance gene of tomato [72]. Cloning and overexpressing the genes responsible for the biocontrol process from Paecilomyces javanicus may reinforce the plant immune response against RKN infection [16]. Likewise, engineered nanomaterials could show promising physical and chemical characteristics against the nematodes [73].” For the reviewer’s question: contrary to Mi1.2, Mi-9 is heat-stable resistance gene. Also, Mi-9 gene was presented in the context of the aforementioned backing references.
Comment: What about the plant-parasitic nematode community. R-genes are nematode specific and its use can lead to the increase of other components of the community. How the molecular plant-nematode interaction can aid with this issue?
Response: As stated in the manuscript, other PPN control methods should work in combination with the molecular approach to solve this issue, as follows: “durability could possibly be maintained via transferring multiple resistance genes to specific cultivar(s) within integrated nematode management systems. In such systems, using crop rotation and/or safe chemical nematicides can assist in reducing pressure on resistant cultivars/varieties to alleviate emergence of virulent populations. BCAs can also offer a significant contribution to at least some of these systems…” These tactics, of course, apply to other components of the community.
Comment: The use of molecular markers can aid in plant breeding for resistance but the response of plants bearing the gene cannot be what be expected, as previously reported. What about that?
Response: Yes, molecular markers can NOT aid in plant breeding for ALL resistance genes hitherto; for few genes only. Therefore, ongoing research are trying to discover more adequate molecular markers for this goal. Sincere thanks again for your constructive comments.
Reviewer 3 Report
The proposed manuscript is a review of a very present day issue in phytonematology and plant pathology, the plant-nematode interaction at cellular and molecular level. It is a very ambitious proposal, and despite demonstrating knowledge on the subject, the author has incurred in several peoblems: (1) the author proposes to review the issue overall, supposedly covering all major PPN (at least that is what can be deduced from the title); however, the review is centered mainly on toot-knot and cyst nematodes, rarely discussing other highly relevant PPN, such as RLN (Pratylenchus spp.), pinewood nematode (Bursaphelenchus xylophilus), to give just two examples; (2) the review is mainly concerned with resistance; perhaps this point should be in the title, and not the broad “plant-nematode interacions”; (3) the English language and writing should be thoroughly reviewed; the author, especially in the first pages frequently uses “colloquial” terms and expressions which are not appropriate in scientific writing; since the manuscript does not come with line numbering, it´s difficult and cumbersome to point out exactly these flaws(4) there is only one figure (Fig. 1) to illustrate some of the main systems described; this would provide some educational value; and even Fig. 1 is not the best way to illustrate the categories; (5) the literature is also not conveniently covered, although it has 128 entries (with 17 self-citations); several authors, which have contributed recently to the topic (e.g. Vieira, Vicente, Espada, etc..) notably on the role and action of effectors or on the ROS response from plants and the mechanisms of nematode overcoming them with associated bacteria (just one example) are not covered; (6) on page 7, point 5.2 the author states that “Nematode reproduction usually undergoes obligate mitotic parthenogenesis…”; this is not true for nematodes in general (which is what the beginning of the phrase suggests) , but only for the cited Meloidogyne species; also in 5.4: the first 5-6 lines apply only ro RKN, so no generalization can be made (“It is well established that the nematode J2…”); (7) the organization of chapters/ points is also confusing; e.g. 7.3 has already been discussed, at least partially in 5.4; so it´s a matter of ordering and organizing the subjects; (8): in 7.4: what species is Galanthus nivalis? In any case , these comments do not demerite the contents which are important and up to date, however, poorly organized. It is suggested for the author to review and resubmit.
Author Response
We thank the reviewer for his/her comments which will improve the manuscript. The responses to all the concerns that were raised are given below. We used the track system with another word color to indicate our complying with his/her comments in a supplementary manuscript.
Comment: The proposed manuscript is a review of a very present day issue in phytonematology and plant pathology, the plant-nematode interaction at cellular and molecular level. It is a very ambitious proposal, and despite demonstrating knowledge on the subject, the author has incurred in several peoblems: (1) the author proposes to review the issue overall, supposedly covering all major PPN (at least that is what can be deduced from the title); however, the review is centered mainly on toot-knot and cyst nematodes, rarely discussing other highly relevant PPN, such as RLN (Pratylenchus spp.), pinewood nematode (Bursaphelenchus xylophilus), to give just two examples;
Response: The most economically important genera: root-knot and cyst nematodes (Meloidogyne, Heterodera, Globodera) were addressed, each genus includes many nematode species that are widespread worldwide. We also mentioned other very important nematode genera related to the core of the research, such as citrus (Tylenchulus), lesion (Pratylenchus), and pine (Bursaphelenchus) nematodes. So I think the title is appropriate, as it is not possible to mention an extremely great number of plant-parasitic nematodes. More importantly, plant-parasitic nematode management or control mostly focuses on these genera due to their economic importance and worldwide spread. The goal of this manuscript is to address molecular approaches for nematode management/control. So, economically important nematodes are on the top to represent PPNs.
Comment: (2) the review is mainly concerned with resistance; perhaps this point should be in the title, and not the broad “plant-nematode interacions”,
Response: The manuscript addressed not only PPN-plant resistance but also presented advances in molecular techniques such as high-quality genome sequencing, interfering RNA (RNAi) and gene editing, all of these techniques can offer a better grasp of these interactions. Efficient tools and resources emanating from such interactions were highlighted while issues in using them were summarized.
Comment: (3) the English language and writing should be thoroughly reviewed; the author, especially in the first pages frequently uses “colloquial” terms and expressions which are not appropriate in scientific writing; since the manuscript does not come with line numbering, it´s difficult and cumbersome to point out exactly these flaws;
Response: Done. Now the manuscript comes with line numbering.
Comment: (4) there is only one figure (Fig. 1) to illustrate some of the main systems described; this would provide some educational value; and even Fig. 1 is not the best way to illustrate the categories;
Response: Fig. 1 is presented only to denote the main systems described.
Comment: (5) the literature is also not conveniently covered, although it has 128 entries (with 17 self-citations); several authors, which have contributed recently to the topic (e.g. Vieira, Vicente, Espada, etc..) notably on the role and action of effectors or on the ROS response from plants and the mechanisms of nematode overcoming them with associated bacteria (just one example) are not covered;
Response: I am awfully sorry not to include all recent references to the topic (e.g. Vieira, Vicente, Espada, etc..) but this was out of hand. It is always improbable to include all such REF due to voluminous research and literature on the topic.
Comment: (6) on page 7, point 5.2 the author states that “Nematode reproduction usually undergoes obligate mitotic parthenogenesis…”; this is not true for nematodes in general (which is what the beginning of the phrase suggests), but only for the cited Meloidogyne species; also in 5.4: the first 5-6 lines apply only ro RKN, so no generalization can be made (“It is well established that the nematode J2…”);
Response: Thanks. I agree that no generalization can be made. So, it was corrected as follows: “Certain nematode reproduction usually undergoes obligate mitotic parthenogenesis – (i.e., M. javanica, M. incognita, and M. arenaria) in the tropics.” Also, in 5.4: the first 5-6 lines, the sentences were modified to be as follows: “It is well-established that the nematode J2, especially RKN, uses a syringe-like stylet to inject a variety of secretory compounds to modify the metabolism of its susceptible host cells. These compounds/effectors are usually synthesized in the J2-esophageal glands and have main commands in forming the hypertrophied multinucleate cells for nematode feeding. They can adjust numerous aspects in terms of physiology and morphogenesis of the invaded plant cells to eventually conquer defense responses and assure nematode development and reproduction on susceptible plants. Thus, the released sets of RKN compounds are involved in inducing pathogenicity…” Kindly, note that RKN was compared with the cyst nematode H. schachtii too.
Comment: (7) the organization of chapters/ points is also confusing; e.g. 7.3 has already been discussed, at least partially in 5.4; so it´s a matter of ordering and organizing the subjects;
Response: Factually, it is impossible to separate some points because they are tightly interrelated, so they may be briefly referred to again when they are closely interconnected with others. However, each title/subtitle occupies the largest and most important text related to it.
Comment: (8): in 7.4: what species is Galanthus nivalis? In any case , these comments do not demerite the contents which are important and up to date, however, poorly organized. It is suggested for the author to review and resubmit.
Response: Galanthus nivalis, is a plant, common name: snowdrop. It is the most widespread of the 20 species in its genus, Galanthus. Its related part in the text “Overexpression of a Galanthus nivalis agglutinin (GNA)-related lectin driven by cauliflower mosaic virus promoter (CaMV35S) is exploited to offer anti-nematode efficacy in plants such as potato, oilseed rape (Brassica napus), and Arabidopsis concerning CNs, RKNs, and Pratylenchus spp. [28].” Sincere thanks again for your constructive comments.
Reviewer 4 Report
Oveqall the review articvle reads well. I
Author Response
We thank the reviewer for his/her comments which will improve the manuscript. We used the track system with another word color to indicate our complying with his/her comments in a supplementary manuscript.
Round 2
Reviewer 3 Report
Authors have seemingly complied with comments and suggestions